# Ball Milling to Produce Composites Based of Natural Clinoptilolite as a Carrier of Salicylate in Bio-Based PA11

**DOI:** 10.3390/polym11040634

**Published:** 2019-04-07

**Authors:** Valeria Bugatti, Paola Bernardo, Gabriele Clarizia, Gianluca Viscusi, Luigi Vertuccio, Giuliana Gorrasi

**Affiliations:** 1Dipartimento di Ingegneria Industriale, Università di Salerno, via Giovanni Paolo II, 132, 84084 Fisciano (SA), Italy; vbugatti@unisa.it (V.B.); gviscusi@unisa.it (G.V.); lvertuccio@unisa (L.V.); 2Nice Filler s.r.l., via Loggia dei Pisani, 25, 80133 Napoli, Italy; 3Istituto per la Tecnologia delle Membrane, Consiglio Nazionale delle Ricerche (ITM-CNR), via P. Bucci 17/c, 87036 Rende (CS), Italy; p.bernardo@itm.cnr.it (P.B.); g.clarizia@itm.cnr.it (G.C.)

**Keywords:** process‒properties relationships, fabrication of drug delivery systems, high-energy ball milling

## Abstract

Antimicrobial packaging systems are recognized as effective approaches to prolong food shelf life. In this context, Bio-based PA11 loaded with a food-grade zeolite were prepared using ball milling technology in the dry state. Zeolite was filled with sodium salicylate, as an antimicrobial agent, and incorporated into the polymer matrix (~50 wt % of salicylate) at different loadings (up to 10 wt %). Structural characterization and an analysis of the physical properties (thermal, barrier, mechanical) were conducted on the composites’ films and compared with the unfilled PA11. The successful entrapment of the antimicrobial molecule into the zeolite’s cavities was demonstrated by the thermal degradation analysis, showing a delay in the molecule’s degradation. Morphological organization, evaluated using SEM analysis, indicated the homogeneous distribution of the filler within the polymer matrix. The filler improves the thermal stability of PA11 and mechanical properties, also enhancing its barrier properties against CO_2_ and O_2_. The elongated form of the zeolite particles, evaluated through SEM analysis, was used to model the permeability data. The controlled release of salicylate, evaluated as a function of time and found to depend on the filler loading, was analyzed using the Gallagher‒Corrigan model.

## 1. Introduction

Conventional food packaging aims at maintenance of food quality, shelf life extension, and assurance of the safety of the food products. Antimicrobial packaging systems are complex devices that incorporate antimicrobial agents into a polymer matrix, providing enhanced food safety [1,2,3]. Food pathogens, being dangerous to consumers’ health, can be managed with multifunctional bio-based antimicrobial packaging agents that inhibit the growth of targeted microorganisms contaminating foods [4,5,6]. Therefore, antimicrobial agents in the packaging system prevent the growth of microbes by extending the initial lag period in which bacteria prepare for their exponential growth, and decreasing live counts of microorganisms by reducing growth rate [7,8,9]. The controlled release packaging systems are a specific form of active packaging. They are time-release or slow-release packaging of active substances, in which the packaging is a delivery vehicle that releases the active molecules over long periods at a controlled rate to the packaged food, to improve its shelf life [10]. Several works have demonstrated through mathematical modeling that controlled release systems play a significant role in the design of appropriate food packaging systems [11,12,13,14]. Despite the growing interest in complex and diversified packaging systems, the concept of controlled release packaging is still being explored for food. 

Controlled release of active molecules can be achieved by adopting the following main strategies [15]: (i) the active molecule is chemically bonded to the polymer having functional groups; (ii) the active molecule is ionically bonded to an anionic or cationic clay, and then dispersed into the polymer matrix; (iii) the active molecule can be encapsulated into a suitable nanocontainer and dispersed into a polymer matrix. Among the possible containers of molecules with specific activities, the zeolites are a class of silicates with peculiar structures and still not fully explored properties as a carrier of active species. Zeolites have the following general formula, M_x_D_y_ [Al_x+2y_Si_n-(x+2y)_O_2n_] * *m*H_2_O, in which M and D represent monovalent and bivalent cations, respectively. They balance the excess of negative charge due to Al^3+^ for Si^4+^ substitutions in the tetrahedral framework. There is a large variety of natural and synthetic zeolites with different chemical composition, pore structures, and crystal sizes. They can be either containers of active molecules to be released, or absorbers of volatile substances. The incorporation of zeolites into polymer matrices can result in novel composite materials with multiple functionalities. 

Polyamide 11 (PA11) or Nylon 11 is a rare bio-based engineering plastic that is derived from renewable resources (castor plants) and produced by polymerization of 11-amino undecanoic acid. Several properties of PA11 are similar to Polyamide 12 (PA12), but it offers superior thermal and UV resistance, low water absorption, and a lower environmental impact. It displays good impact strength and dimensional stability [16,17,18]. It is thus a promising “green” alternative material in a variety of applications, ranging from structural applications to packaging. 

This paper reports the preparation of composites based on a food-grade zeolite filled with sodium salicylate, as an antimicrobial agent, listed in EC-Directive 10/2011/EC of 14 January 2011, and dispersed in a PA11 matrix with different loadings. The processing window of PA11 is around 200 °C and the sodium salicylate, in the processing conditions of PA11, can undergo thermal degradation. In order to preserve the thermal stability of the active molecule, the methodology used to incorporate the zeolite-salicylate active filler was a solid state mixing through a high-energy ball milling apparatus, in dry conditions at ambient temperature. It has been demonstrated that such methodology could represent an ecological and economical strategy for the fabrication of structural and functional polymeric composites and nanocomposites, avoiding high temperatures and solvents [19,20]. The most important advantages of this technology include the control of the degradation processes associated with high temperature, a strong reduction of environmental disposal, the compatibilization of immiscible blends and the treatment of waste disposal and recycled materials. The dispersion of fillers and nano-fillers into a polymer and the proper manipulation of thermo-sensitive active molecules such as antimicrobials, oxygen scavengers, and antibiotics are other advantages of this process, applied to composites’ manufacturing.

## 2. Materials and Methods

### 2.1. Materials

Zeolite MED® detox ultrafine powder, Medical device, an ultra-fine natural clinoptilolite zeolite powder, was purchased from Zeolith-Bentonit-Versand.de. Sodium salicylate (CAS Number 54-21-7) and PA11 (CAS Number 25035-04-5) were purchased from Sigma Aldrich (Milan, Italy). All materials were used as received. 

### 2.2. Preparation of Zeolite-Salicylate Hybrid Filler

Three grams of sodium salicylate were dissolved in 30 mL of distilled water at 50 °C for 20 min. The zeolite (3 g) was mixed with the sodium salicylate solution and ultrasonicated for 10 min, at 40% amplitude, using a UP200S Ultrasonic Processor (Heilscher, Teltow, Germany), in order to allow the dispersion of the zeolite in the sodium salicylate solution. The solution was heated at 50 °C for 2 min, then reduced pressure (0.085 MPa) was applied to remove the air between and within the holes. The vacuum was maintained for 15 min. The solution was then taken out of the vacuum and shaken for 5 min. Then a vacuum was re-applied to remove the trapped air for 15 min. The zeolite loaded with salicylate (Zeo-sal) was filtered and dried in an oven for 16 h at 50 °C up to a constant weight. The vacuum cycle was then repeated a second time. 

### 2.3. Incorporation of Zeolite-Salicylate into the PA11 Matrix

The incorporation of the zeolite‒salicylate nano-hybrids into PA11 was achieved by the high-energy ball milling (HEBM) method. Powder mixtures composed of PA11 and 3, 5, 7, and 10 wt % of zeolite-salicylate (vacuum dried for 24 h) were milled at room temperature in a planetary ball mill (Retsch, model PM 100 (Haan, Germany)). The milling process occurred in a cylindrical steel jar of 50 cm^3^ with five steel balls of 10 mm diameter. The rotation speed was 580 rpm and the milling time was 60 min. Pure PA11, taken as a reference, was milled in the same experimental conditions used for the composites. The milled powders were molded in a Carver laboratory press between two Teflon sheets, at 190 °C, followed by cooling at ambient temperature. Films of thickness of about 150 μm were obtained and analyzed.

### 2.4. Characterization

X-ray diffraction (XRD) patterns were taken, in reflection, with an automatic Bruker (Milano, Italy) diffractometer (equipped with a continuous scan attachment and a proportional counter), using nickel-filtered Cu Kα radiation (Kα = 1.54050 Å) and operating at 40 kV and 40 mA, step scan 0.05° of 2ϑ and 3 s of counting time.

Thermogravimetric analyses (TGA) were carried out in air atmosphere with a Mettler (Novate Milanese Milan, Italy) TC-10 thermobalance from 30 to 800 °C, at a heating rate of 10 °C/min.

Fourier transform infrared (FT-IR) absorption spectra were recorded by a Bruker spectrometer (Bruker Italia, Milano, Italy), model Vertex 70 (average of 32 scans, at a resolution of 4 cm^−1^).

Scanning electron microscopy (SEM) was adopted to investigate the morphology of the films. Before the analysis, the film samples were fractured in liquid nitrogen in order to observe the cross section without altering the original structure. Both films and zeolite powder were coated with a thin film of gold by sputtering. Images were acquired by an EVO|MA 10 (Zeiss, Milan, Italy) microscope, working in high-vacuum mode. 

The mechanical properties of the samples were evaluated at room temperature using a dynamometric apparatus INSTRON 4301 (Coronation Road, High Wycombe, Buckinghamshire, HP12 3SY, UK). Experiments were conducted on samples (10 mm × 5 mm × 0.150 mm) in tensile mode with a deformation rate of 10 mm/min. The initial length of the samples was 10 mm. The elastic modulus, *E* (MPa), was calculated from the linear part of the stress-strain curves, which extended to a deformation of 0.2%. Five samples for each film were tested and standard deviation calculated.

Gas permeation tests on neat and filled PA11 samples were carried out with single gases (O_2_ and CO_2_) at a feed pressure of 1 bar and 25 °C. A fixed volume/pressure increase instrument (Elektro & Elektronik Service Reuter, Geesthacht, Germany), described elsewhere [21], was used. Before each experiment, the membrane samples were thoroughly evacuated using a membrane pump for the pre-vacuum and an oil-free turbo-molecular pump for a high vacuum. A pressure transducer in a calibrated volume at the permeate side monitored the pressure increase over time starting from the membrane exposure to the feed gas. The gas permeability (*P*) was obtained from the slope of the pressure curve at steady state conditions and is expressed in Barrer (1 Barrer = 10^−10^ cm^3^ (STP) cm cm^−2^ cmHg^−1^·s^−1^). The extrapolation of the linear portion of the pressure curve on the abscissa provides the gas time lag (*θ*) that can be used to evaluate the diffusion coefficient, *D*, of each gas through the membrane [22]:
*D* = *l*^2^ /6 *θ*(1)
where *l* is the membrane thickness.

The ‘solution-diffusion’ transport model (*P* = *D S*), typically assumed in dense polymeric films [23], was applied for discussing the gas transport properties of the membranes. Accordingly, the solubility coefficient (*S*) can be indirectly obtained from the permeability and diffusion coefficient. 

The ideal selectivity was calculated as the ratio of the individual permeability values for the two gases:
α_CO_2_/O_2__ = *P*_CO_2__/*P*_O_2__.
(2)

Specimens with a circular effective area of 11.3 or 2.14 cm^2^ were used in permeation tests. Their thickness was measured with a digital micrometer (IP65, Mitutoyo, Milan, Italy), taking the average of multiple point measurements. Three samples for each film were tested and standard deviation calculated.

### 2.5. Gas Transport Modeling

For a mixed matrix membrane (MMM), the gas permeability can be estimated by different theoretical models. The Maxwell model [24] predicts the apparent gas permeability of the MMMs (*P_MMM_*) by combining the permeability of the continuous matrix (*P*m) and of the dispersed phase (*P*d) with the volume fraction of the dispersed phase (*φ*d):(3)$$P_{MMM}=P_{m}\cdot \frac{P_{d}+2P_{m}-2\phi_{d}\left(P_{m}-P_{d}\right)}{P_{d}+2P_{m}+\phi_{d}\left(P_{m}-P_{d}\right)}.$$

A modified version of the Maxwell model (Maxwell‒Wagner‒Sillar) was proposed by Bouma et al. in the case of heterogeneous membranes [25]. It contains an additional parameter, *n*, the shape factor of the particles:(4)$$P_{MMM}=P_{m}\cdot \frac{n\;P_{d}\;\left(1-n\right)P_{m}-\left(1-n\right)\phi_{d}\left(P_{m}-P_{d}\right)}{n\;P_{d}+\left(1-n\right)P_{m}+n\phi_{d}\left(P_{m}-P_{d}\right)}.$$

This model can be applied to represent the dispersed particles as ellipsoids, using a shape factor 0 < *n* < 1/3 for prolate ellipsoids (longest axis along the direction of the applied pressure gradient across the membrane) and a shape factor in the range 1/3 < *n* < 1 for oblate ellipsoids (shortest axis in the direction of applied pressure gradient). In the case of spherical particles (*n* = 1/3), the model reduces to the Maxwell estimation.

In order to convert the filler mass fractions in volume fractions, the following density data were used: Zeolite density = 2.18 g·cm^−3^, PA11 density = 1.026 g·cm^−3^. 

The release kinetics of the salicylate was tested on the PA11/Zeo-sal films, using rectangular specimens of 4 cm^2^ and same thickness (200 μm), placed into 25 mL of ethanol and stirred at 100 rpm in an orbital shaker (VDRL MOD. 711+, Asal S.r.l., Florence, Italy). The release medium was withdrawn at fixed time intervals and replenished with fresh medium. The salicylate release was monitored using a UV-2401 PC spectrometer (Shimadzu, Tokyo, Japan). The considered band was at 230 nm. 

## 3. Results

### 3.1. Characterization of Zeolite and Zeolite-Salicylate Hybrids

The morphological analysis on the Zeo-sal powder (Figure 1) evidenced an elongated form (platelets having the larger dimension of ca. 1 micron). 

The TGA analysis carried out on the salicylate, zeolite, and zeolite-salicylate hybrid powder provided the curves reported in Figure 2 for the weight loss percent and its derivate (DTG). The small weight variation of pure zeolite is due to the loss of water molecules inside the zeolite’s structure. The sodium salicylate shows two main weight losses (Figure 2b), at 275 and 343 °C, respectively. The first decomposition step is attributed to a decarboxylation process, leading to carbon dioxide, phenoxide and phenol. The second event is due to the formation of benzene, low molecular weight oxygenated compounds, a mixture of stable sodium salts such as oxide, hydroxide, or carbonate [26]. The second decomposition step results delayed of about 20 °C for the zeolite-salicylate hybrid. This could be an indication of the occurred entrapment of a part of salicylate into the zeolite channels. 

The content of salicylate (wt %) in the zeolite-salicylate hybrid has been calculated using the TGA analysis according to the following equation:
*α*_3_ = *w**α*_1_ + (1 − *w*) *α*_2_(5)
where *α*_1_ is the mass loss of salicylate (67.15%) at 289.17 °C; *α*_2_ is the mass loss of zeolite (10.25%) at 141.91 °C; *α*_3_ is the mass loss of zeolite-salicylate (43.49%) at 278.32 °C.

Therefore, the content of salicylate in the zeolite-salicylate hybrids was estimated to be 51.43 wt %.

### 3.2. Characterization of PA11 and PA11 Composites

SEM images taken on the PA11 films are shown in Figure 3. The PA11 film with the lowest concentration (3%) shows a not perfectly smooth surface (Figure 3a). The composite films with the highest concentration (10%) showed zeolite particles (size of about 0.5 microns) covered by a polymer layer (Figure 3b). The fractured cross section of the PA11 + 10% Zeo-sal film displays a quite good distribution of fillers (Figure 3c). A larger filler amount is visible at one side of the film surfaces (filler size of about 0.7 micron), probably due to sedimentation of the filler itself (Figure 3d).

Figure 4 reports the XRD spectra for the composite materials and for the unfilled PA11 that was processed in the same conditions of the composites. The XRD spectrum of PA11 is typical of the triclinic α form, characterized by reflections at the diffraction angle 2θ ≈ 7.5°, 2θ ≈ 20.5° and 2θ ≈ 23.2° corresponding to (001), (100) and (010)/(110) planes, respectively [27]. Spectra of composites show that PA11 crystallizes in the same triclinic α form in presence of the filler. The spectrum of zeolite shows the peak at 2θ ≈ 9.80° typical of the considered clinoptilolite (*d* = 8.950 Å), the other two characteristic peaks of the clinoptilolite, at 2θ ≈ 22.4° (*d* = 3.955 Å) and 2θ ≈ 30.0° (*d* = 2.971 Å) [28], are hidden in the composites from the characteristic peaks of the polymer matrix.

Figure 5 reports the TGA thermograms (a) and DTG (b) overlays of the neat PA11 and PA11 composites. The thermal degradation of PA11 displays two steps, similarly to other polyamides [29]. The first step of degradation, located around 430 °C, generates gaseous species containing C=O, N–H and CH_2_ groups [30,31,32], mainly lactams, nitriles, and unsaturated hydrocarbons. It is assumed that these decomposition products are released in this temperature range. During the second step of decomposition, at around 470 °C, the products are mainly constituted by hydrocarbons and oxygenated low molecular weight organic compounds. The presence of the Zeo-Sal filler delayed the degradation of PA11, up to 40 °C for the first degradation step and up to 50 °C for the second one (composites with 7 and 10 wt % of zeolite-salicylate). This could be due to either to interactions between filler and polar groups of the polymer matrix, i.e., NH and C=O groups or to the slower diffusion of CO_2_, as product of decomposition (see barrier properties discussion). Composites with 3 and 5 wt % of Zeo-Sal show an intermediate behavior between the unfilled polymer and the composite at the highest loading. It is interesting to observe that no differences in the thermal degradation are evident for filler loadings larger than 7 wt %.

Figure 6 reports the FTIR spectra of filler and composite membranes of PA11. The zeolite spectrum displays characteristic peaks at 1634 cm^−1^ for the vibration of the Si‒O bond and at ca. 1070 cm^−1^ for the vibrations generated by Al‒O bonds. In the 1700–2000 cm^−1^ region, the so-called benzene fingers are observed for the salicylate [33]. The Zeo-Sal hybrid shows characteristic peaks for the salicylate (Figure 6a). The phenolic group of sodium salicylate provide peaks at ca. 1370 cm^−1^ indicating the bending mode (Ph-OH) and at ca. 1250 cm^−1^ from Ph-O. The peaks at ca. 1580 and 1485 cm^−1^ are assigned to the symmetric and asymmetric carboxylate (COO) bands. 

The neat PA11 presents intense bands close to those typically found at 1634 cm^−1^ (C=O stretching) and 1540 cm^−1^ (C–N stretching + N–H bending). The double peak (721 and 687 cm^−1^) is associated to the crystalline γ phase [34,35]. The bands of PA11 are clearly visible in the composites (Figure 6b). In addition, in the region 700‒800 cm^−1^, the composites display specific peaks evidenced in the Zeo-Sal hybrid (e.g., the peak at 755 cm^−1^ indicating the out-of-plane C‒H bending aromatic [36], Figure 6c). In general, no relevant changes are present in the composite PA11-based materials.

Figure 7 reports the elastic modulus, *E* (MPa), as function of filler loading. It is evident a significant increasing of the elastic modulus up to 5 wt % of filler. After such percentage, the elastic modulus tends to reach a plateau. Indeed, also the TGA analysis evidenced a comparable behavior for the samples containing 7% and 10% of the Zeo-Sal. The increase in the modulus is lower than a linear trend that could be expected assuming a perfect adhesion between the polymer and the fillers. The observed trend is an indication of agglomeration of the fillers as their loading increases, as evidenced by SEM images.

### 3.3. Gas Permeation on Films Based on PA11 and PA11 Composites

The gas permeation tests on the PA11-based films provided the following transport parameters: permeability, diffusion and ideal selectivity (Figure 8). CO*_2_* was the most permeable species, also in the filled samples. In the presence of larger voids, the Knudsen diffusion would favor the O_2_ permeation over CO_2_ diffusion. Therefore, the data suggest a good interfacial adhesion between the zeolites and PA11 and the absence of large cavities that would function as bypasses around the filler particles. These defects, resulting in larger permeability and in a selectivity loss, would deteriorate the barrier performance of the film. Despite the porous nature of the incorporated zeolites, a permeability reduction was observed with respect to the pristine PA11. The addition of Zeo-Sal to the PA11 matrix progressively reduced the gas permeability of the two gases, keeping almost constant the CO_2_/O_2_ permselectivity. The gas permeability reduction of the whole composite system could be related to the polymer chains obstructing the access to the filler voids, or to clogged zeolite particles owing to the entrapped salicylate molecules. 

An analysis of the diffusion and solubility coefficients shows distinct trends for the two gases. The tendency of CO_2_ to interact with the functionalized zeolites leads to a reduction in its diffusion coefficient, whereas for the less interactive species as the oxygen, owing to additional paths for the diffusion, the behavior is opposite. In particular, the reduction in the apparent diffusion coefficient of CO_2_ is typically found in zeolite-based nanocomposite films and described as immobilizing adsorption of diffusing penetrants [37]. Usually, it causes very large increases in the diffusion time lag, but has negligible effects on the steady-state permeation rate. In addition, the calculated solubility parameter of CO_2_ is larger than in the neat PA11 as the filler concentration increases, indicating a preferential affinity of the zeolite for this highly sorbing species. This combined effect on solubility and diffusivity could be responsible for a reduced CO_2_ permeation rate [38]. On the contrary, a larger apparent diffusion coefficient in the MMMs with respect to the neat polymer and a lower solubility were observed for O_2_ upon the zeolite addition to PA11.

The gas permeability through the composite films was interpreted with different models. The Maxwell model was adopted considering fillers with larger permeability than the polymer matrix (*P*d ≫ *P*m) or no permeability (*P*d = 0) as upper and lower limit, respectively. This approach is generally applied for diluted systems, as in the present case (filler concentration < 20 vol %) and non-interacting spherical particles. In the case of CO_2_ permeability, the experimental data are lower than the prediction obtained using the Maxwell model for impermeable particles (Figure 9). SEM images showed a non-spherical shape for the functionalized zeolite powder (Figure 1). Therefore, we considered the Maxwell‒Wagner‒Sillar model to predict the gas transport behavior of the zeolite-based MMMs considering impermeable fillers and taking into account the effect of the particle shape. A good agreement between the experimental and modeled results was obtained when using a shape factor of 1/1.24 as a fitting parameter. In this case, the system could be modeled considering oblate ellipsoids (shape factor in the range 1/3 < *n* < 1) that act as a barrier. Owing to the alignment of the crystals in the film (oblate ellipsoids), the tortuosity is higher and permeability decreases. Therefore, filler particles having a larger shape factor are more effective than spherical particles in decreasing the gas permeability of a fixed polymer. On the basis of these results, a larger amount of Zeo-Sal should result in a more tortuous diffusion pathway of the drug through the PA11 polymeric matrix.

### 3.4. Release Kinetics of Salicylate

Figure 10 reports the release of salicylate for the films with different filler loadings, as function of time (up to 250 h). Two steps are evident in the release curves: a first one attributed to the burst of the salicylate molecules located on the external surface of the composites and a second one related to the counter-diffusion of the molecule from the bulk. After the second step, it is possible to observe a plateau. 

It is worth to note that the amount of released salicylate molecule decreases with increasing the filler loading. The release is almost complete in the case of PA11 + 3% Zeo-Sal. In the other cases, the plateau is below 70%. The gas permeation tests evidenced a reduction for the diffusion coefficient for these small molecules as the loading of the filler increases in the films. The modelling indicated that Zeo-Sal behave as impermeable obstacles. Indeed, the tested gases can be seen as probes of the film microstructure. Therefore, the diffusion of salicylate is more hindered in the films presenting a larger amount of filler. 

In order to fit the experimental data through phenomenological interpretation, the Gallagher and Corrigan model was used [11]. It describes the kinetics of drug release from a polymeric system in two stages: the first part of the equation reflects the diffusion-controlled dissolution of the drug into the medium, which is characterized by first-order kinetics; the second part reports that the drug release rate depends on the polymer relaxation [11,12,38]. Therefore, *f*_t_, the accumulative drug release percentage at time *t*, is given by:(6)$$f_{t}=f_{b}*\left(1-e^{-k_{1}t}\right)+\left(f_{tmax}-f_{b}\right)\left(\frac{e^{k_{2}\left(t-t_{2max}\right)}}{1+e^{k_{2}\left(t-t_{2max}\right)}}\right),$$
where *k*_1_ is the first order release constant (Stage 1), *k*_2_ is the second stage release constant due to the polymer relaxation, *f*_b_ is the accumulative drug release percentage during the Stage 1, *f*_tmax_ is the maximum drug release percentage during the whole process, and *t*_2max_ is the time at which the drug release rate reaches the maximum. 

Table 1 reports the kinetic parameters derived from Figure 10 using Equation (6). The burst parameter (*f*_b_) decreases with filler loading. This fraction of the drug was the most accessible one, being physically adsorbed to the external surface of the zeolite. This phenomenon could be related either to a barrier effect created by the zeolite located on the composite’s surface and/or to a weak interaction between salicylate molecules with the polymer matrix (confirmed by FTIR). The *k*_1_ and *k*_2_ constants do not change significantly, while the time at which the drug release rate reaches the maximum, *t*_2max_, increases. Therefore, the drug diffusion behavior, especially the early time release, can be manipulated by changing the drug loading and consequently the drug diffusivities in the composite assembly.

## 4. Conclusions

This paper reports the preparation and characterization of novel composites based on commercial bio-based PA11 matrix filled with nano-hybrids composed of a food-grade zeolite (clinoptilite) and sodium salicylate, as an antimicrobial agent. The potential applications of the considered composites are in the food packaging field. The technology used for the incorporation of the fillers into the polymer matrix was high-energy ball milling at ambient temperature and in dry conditions. The delayed decomposition step for the zeolite-salicylate hybrid suggested a good entrapment of a part of salicylate into the zeolite channels. The elastic modulus resulted improved with the filler loading, especially at low concentration. At the same time, the filler improves the thermal stability of PA11, leading to composite films with enhanced barrier properties when tested in gas permeation for CO_2_ and O_2_. SEM analysis indicated a good distribution of the fillers within the PA11 matrix, evidencing an elongated form for the zeolite particles. These morphology details were used to model the permeability as a function of the filler concentration, considering the fillers as impermeable particles. The release of salicylate from composites’ membranes was found to be strictly dependent on the filler loading and was well fitted by the Gallagher‒Corrigan model. It was demonstrated that by varying the filler loading it is possible to tune the rate of release of the entrapped salicylate molecule for targeted applications.

## Figures and Tables

**Figure 1 polymers-11-00634-f001:**
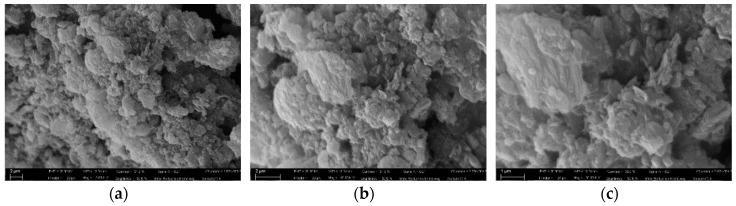
SEM images of the Zeolite-salycilate powder taken at different magnifications. (**a**) 8000X; (**b**) 16,000X; (**c**) 30,000X.

**Figure 2 polymers-11-00634-f002:**
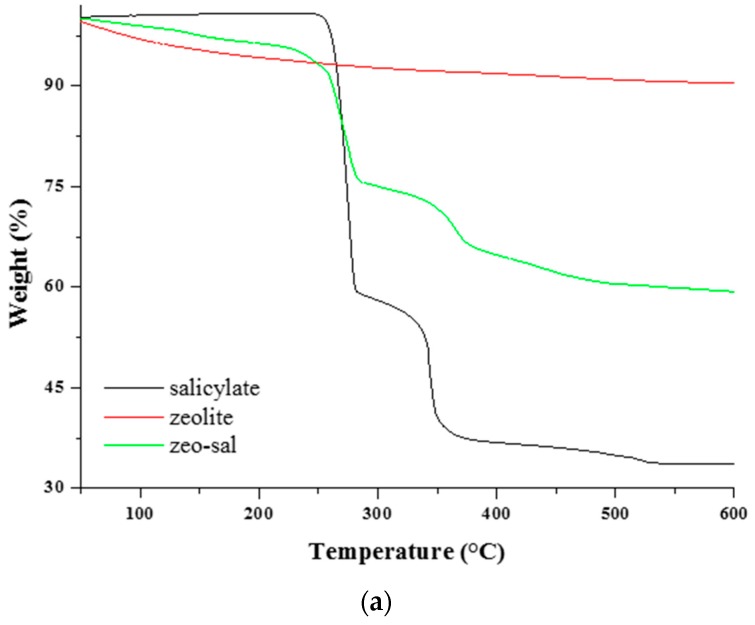
Thermogravimetric analysis on the salicylate, zeolite, and zeolite-salicylate hybrid. (**a**) TGA curves; (**b**) DTG curves.

**Figure 3 polymers-11-00634-f003:**
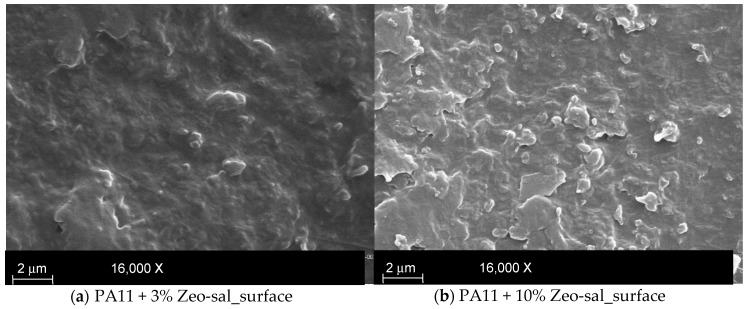
SEM images of the films. (**a**) surface of the PA11 film PA11 + 3% Zeo-sal; (**b**) surface of the PA11 + 10% Zeo-sal film; (**c**) cross section of the film PA11+ 10% Zeo-sal film; (**d**) detail of the cross section of the film PA11+ 10% Zeo-sal film di PA11 + 10%.

**Figure 4 polymers-11-00634-f004:**
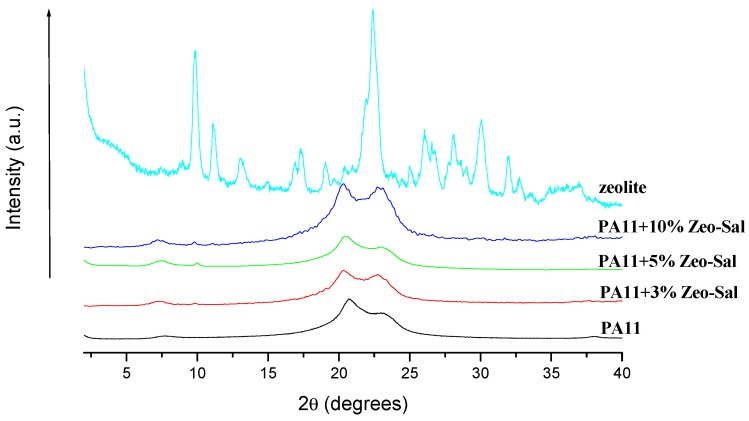
XRD spectra of: zeolite, unfilled PA11, and composite materials (PA11 + Zeo-Sal).

**Figure 5 polymers-11-00634-f005:**
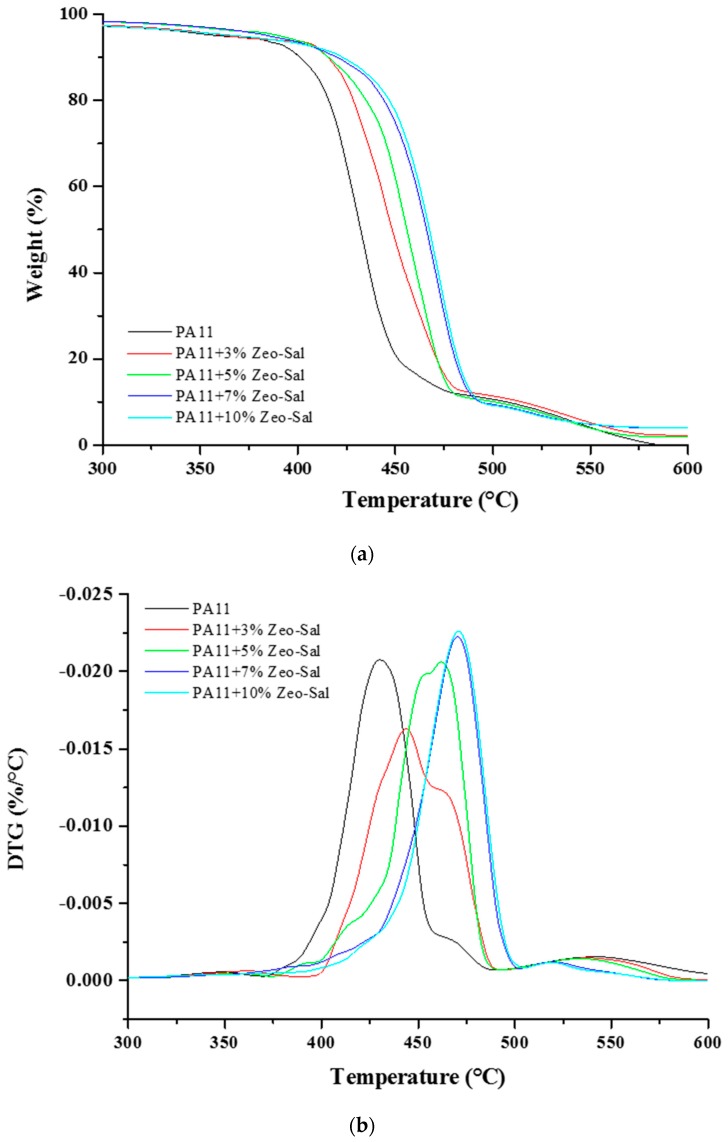
Thermogravimetric analysis on PA11 and composite materials (PA11 + Zeo-Sal). (**a**) TGA curves; (**b**) DTG curves.

**Figure 6 polymers-11-00634-f006:**
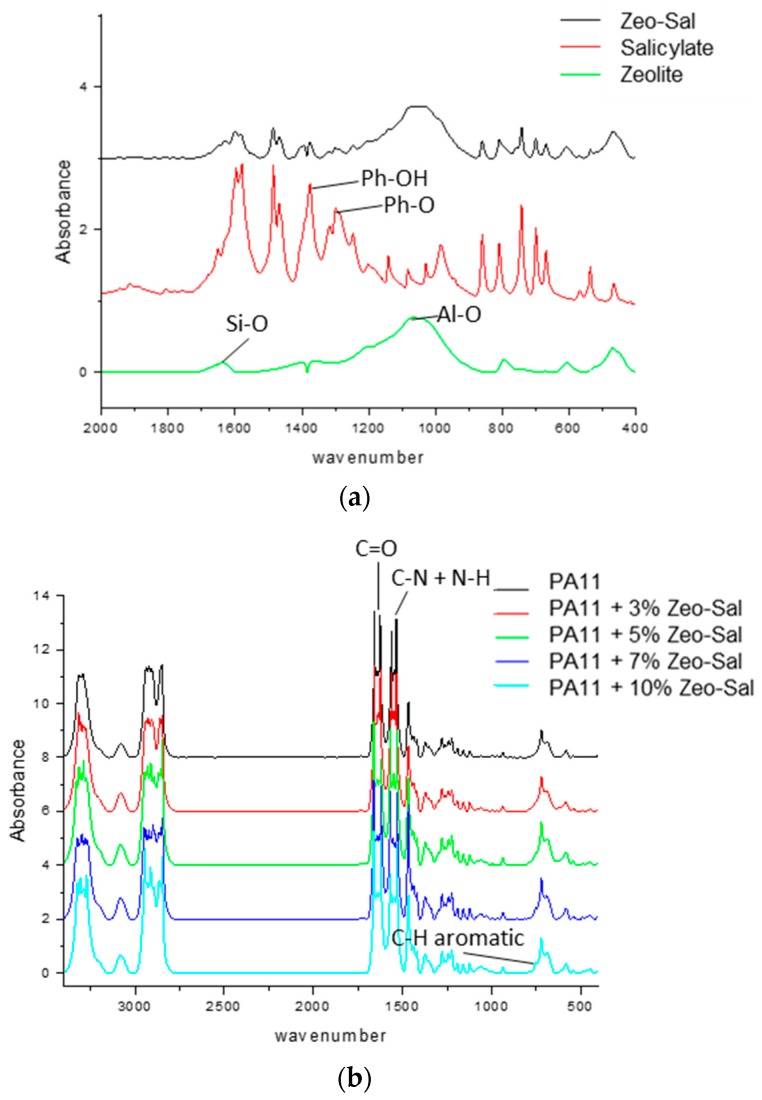
FTIR spectra. (**a**) Salicylate, zeolite, and zeolite-salicylate hybrid; (**b**,**c**) unfilled PA11 and composite materials (PA11 + Zeo-Sal).

**Figure 7 polymers-11-00634-f007:**
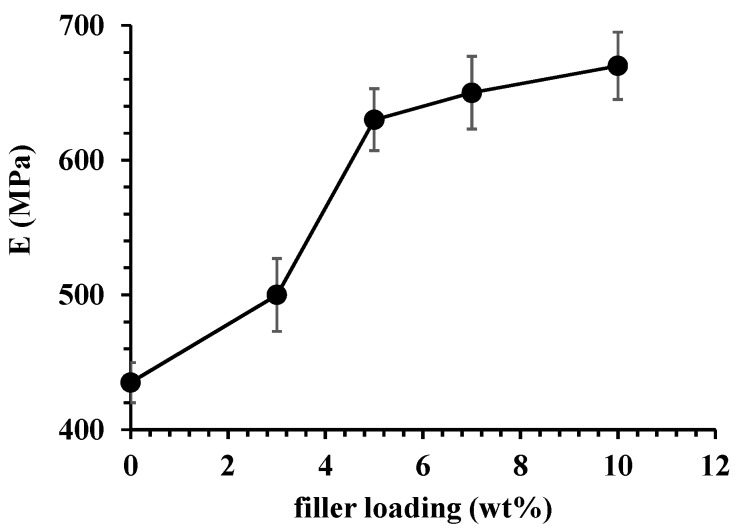
Elastic modulus as function of filler loading for composite materials (PA11 + Zeo-Sal).

**Figure 8 polymers-11-00634-f008:**
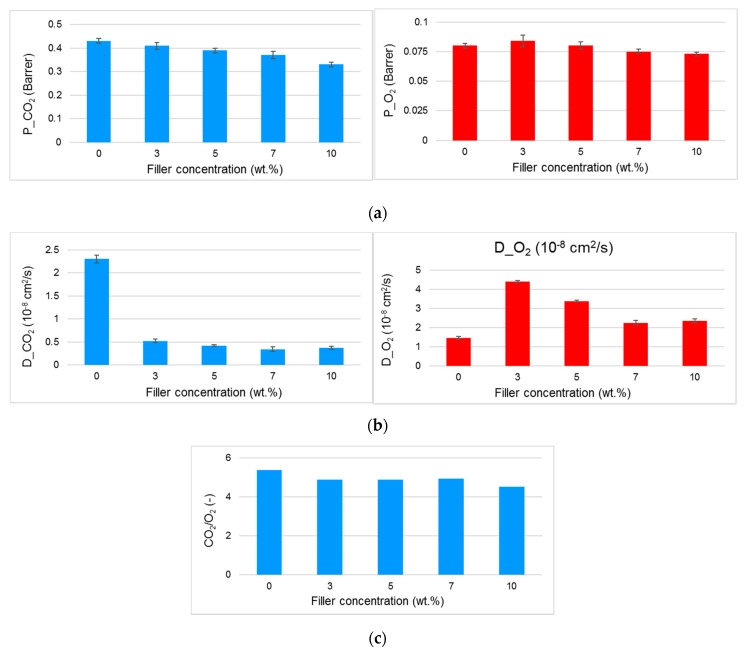
Transport parameters for CO_2_ and O_2_ measured at 25 °C on films based on PA11 and composite materials (PA11 + Zeo-Sal). (**a**) Permeability; (**b**) diffusion coefficient; (**c**) permselectivity.

**Figure 9 polymers-11-00634-f009:**
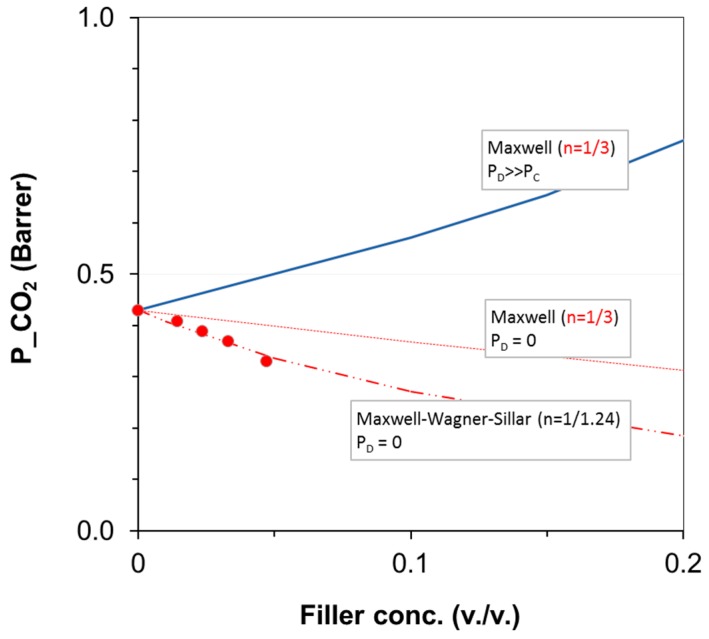
CO_2_ permeability measured through films of PA11 loaded with Zeo-Sal as a function of the filler volume fraction. The continuous lines correspond to the upper limit (*P*d ≫ *P*m) and lower limit (*P*d = 0) of the Maxwell model. The dashed line corresponds to the fitting of the experimental points using the Maxwell-Wagner-Sillar model with *P*d = 0.

**Figure 10 polymers-11-00634-f010:**
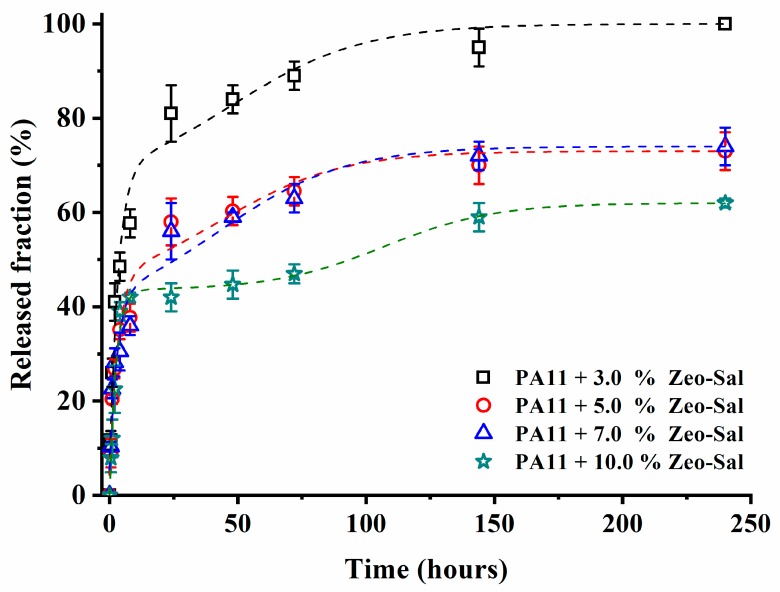
Released fraction of salicylate as function of time for films of PA11 filled with different Zeo-Sal loadings.

**Table 1 polymers-11-00634-t001:** Kinetic parameters for the salicylate release derived from Equation (5).

Filler loading (wt %)	*f*_b_ (%)	*t*_2max_ (h)	*k*_1_ (h^−1^)	*k*_2_ (h^−1^)
3.0	65	47	3.03E-01	3.89E-02
5.0	43	43	3.09E-01	4.23E-02
7.0	38	42	3.43E-01	4.09E-02
10.0	44	107	4.02E-01	4.65E-02

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
