# Peer review of "Ball Milling to Produce Composites Based of Natural Clinoptilolite as a Carrier of Salicylate in Bio-Based PA11"

_polymers, 2019, doi:10.3390/polym11040634_

Round 1

Reviewer 1 Report

The manuscript entitled "Ball Milling to produce composites based of natural clinoptilolite as carrier of salicylate in bio-based PA11", by Bugatti and colleagues, looks into the application of high-energy ball milling technique to manufacture PA11 polymer nanocomposite films. This is a technical article, delineating the detailed steps of manufacturing and characterization for a potentially valuable film material with applications in controlled release packaging. While great technical data are smoothly provided, there seem to be a number of minor issues that could be addressed to significantly enhance the impact and scope of the work. Below I have listed the specific comments for the authors:

The Abstract needs major improvement. What is written is a summary of methods used, with no description of introduction (what the problem is, significance, etc.), obtained RESULTS and ultimate/potential outcomes. This current version of Abstract would not engage readers to read the article. 

Figure 1: a single panel of SEM image seems insufficient to make a figure for this manuscript. I suggest authors to add multiple SEM images, at varying magnifications and demonstrating different features of the material.

Figure 3: these SEM images do not provide much info in the current form. Too dark, with little features clear in these images. Images are taken at different magnifications and direct comparison is not advisable. I recommend providing same exact magnifications for all panels and highlighting the specific features in the ultrastructures that you'd like reader to pay attention to (using arrows).

Figures 4 and 5: best would be to combine these small figures and make a multi-panel, more standard format of a figure.

Figure 7: I strongly suggest major modifications to be applied to the graphs presented in this figure: Y axis with proper labeling should be added, error bars are missing, and "n" (sample size) and statistics (P value) are not reported.

Author Response

Reviewer #1

The manuscript entitled "Ball Milling to produce composites based of natural clinoptilolite as carrier of salicylate in bio-based PA11", by Bugatti and colleagues, looks into the application of high-energy ball milling technique to manufacture PA11 polymer nanocomposite films. This is a technical article, delineating the detailed steps of manufacturing and characterization for a potentially valuable film material with applications in controlled release packaging. While great technical data are smoothly provided, there seem to be a number of minor issues that could be addressed to significantly enhance the impact and scope of the work.

We thank the Reviewer for his/her appreciation to the work.

Below I have listed the specific comments for the authors:

The Abstract needs major improvement. What is written is a summary of methods used, with no description of introduction (what the problem is, significance, etc.), obtained RESULTS and ultimate/potential outcomes. This current version of Abstract would not engage readers to read the article. 

Author reply à We completely rewrote the Abstract highlighting the context and the main results.

Figure 1: a single panel of SEM image seems insufficient to make a figure for this manuscript. I suggest authors to add multiple SEM images, at varying magnifications and demonstrating different features of the material.

Author reply à Novel images are provided for the Zeosal powder (Fig. 1) using different magnifications (8000X Fig. 1, a, 16000X Fig. 1 b and 30000X Fig. 1 c) (see below).

a)       8000X

b)       16000X

c)       30000X

Figure 3: these SEM images do not provide much info in the current form. Too dark, with little features clear in these images. Images are taken at different magnifications and direct comparison is not advisable. I recommend providing same exact magnifications for all panels and highlighting the specific features in the ultrastructures that you'd like reader to pay attention to (using arrows). 

Author reply à The SEM images included in the manuscript as Fig. 3 show two surfaces of the neat PA11 and of a PA11+Zeosal sample (Fig. 3, a and b, respectively). They were taken at the same magnification (16000 X).

Novel images are used (Fig. 3, c and d) with a better resolution and brightness (see below). They show the cross-section of the PA11+Zeosal sample using different magnifications in order to evidence the whole section (Fig. 3 c, 2400X) and a zoom on the particles incorporated in the PA11 (Fig. 3 d, 10000X).

(c)           PA11 +10%   Zeo-sal_cross-section

(d) PA11 +10% Zeo-sal_detail of the cross-section

Figures 4 and 5: best would be to combine these small figures and make a multi-panel, more standard format of a figure. 

Author reply à Figure 4 is relative to XRD analysis on the considered samples (structural analysis) and Figure 5 is relative to the thermogravimetric analysis (TGA and DTG) (physical property: thermal degradation). We prefer to leave separately such analyses, in order to better discuss the physical properties and their correlation to the structural organization of the composites.

Figure 7: I strongly suggest major modifications to be applied to the graphs presented in this figure: Y axis with proper labeling should be added, error bars are missing, and "n" (sample size) and statistics (P value) are not reported. 

Author reply à The permeability tests were done on three samples for each concentration (now specified in the text). Novel figures (now Fig. 8, a, b and c) were added for the transport parameters in which the data represent the average, with error bars related to the standard deviations.

We add the revised manuscript with highlighted in yellow our corrections/modifications according to Reviewer's suggestions

Reviewer 2 Report

This manuscript report preparation composite film based on the polyamide 11 polymer matrix and food grade zeolite. This work is interesting, but it needs significant revision before its publication in Polymers.

1.          In Abstract more result should be added.

2.          Introduction need revision and more up to date references should be added.

3.          The results and discussion part should be improved. The author should add more up to date reference and compare the obtained results with available literature.

4.           Resolution of Figures are not publishable level, revise it carefully. In Figure 3 the scale bar of each image is different and hence difficult to compare the results. In revised manuscript, author should add all the SEM Figure in same scale bar. In Figure 4 inset is not clear. In Figure 6 (b-d) is also not clear, why 3 different FTIR image was used. Author should add a FTIR of 400-3400 Cm-1 range and also define all the FTIR peak in Figure.

5.         What about the biocompatibility of the developed composite material?

6.        What is the application of the prepared composite?  Here sodium salicylate was used as an antimicrobial agent so where is antimicrobial activity test results?

7.       What about the mechanical properties of the prepared composite film?

8.         The discussion is too short (Line 311), expand it.

9.       In line 311, Author claimed nanocomposite but the results do not support the same. Give a proper explanation.  

10.         The conclusion is missing, add a brief conclusion at the end of the manuscript.

11.       Also, carefully revise the manuscript for typographical and linguistic error.

Author Response

Reviewer #2

This manuscript report preparation composite film based on the polyamide 11 polymer matrix and food grade zeolite. This work is interesting, but it needs significant revision before its publication in Polymers.

We thank the Reviewer for his/her appreciation to the work.

1.                  In Abstract more result should be added.

Author reply à We completely rewrote the abstract highlighting the main results.

2.                  Introduction need revision and more up to date references should be added.

Author reply à We revised the introduction and we have added the most recent references on the topic.

3.                  The results and discussion part should be improved. The author should add more up to date reference and compare the obtained results with available literature.

Author reply à We revised the discussion part adding the most recent references on the topic.

4.             Resolution of Figures are not publishable level, revise it carefully.

In Figure 3 the scale bar of each image is different and hence difficult to compare the results. In revised manuscript, author should add all the SEM Figure in same scale bar.

Author reply à  The SEM images included in the manuscript as Fig. 3 show two surfaces of the neat PA11 and of a PA11+Zeosal sample (Fig. 3, a and b, respectively). They were taken at the same magnification (16000 X).

Novel images are used with a better resolution and brightness (see below, Fig. 3, c and d). They show the cross-section of the PA11+Zeosal sample using different magnifications in order to evidence the whole section (Fig. 3 c, 2400X) and a zoom on the particles incorporated in the PA11 (Fig. 3 d, 10000X).

(c)           PA11 +10%   Zeo-sal_cross-section

(d) PA11 +10% Zeo-sal_detail of the cross-section

In Figure 4 inset is not clear.

Author reply à We introduced into the figure the spectrum of zeolite, removing the inset.

In Figure 6 (b-d) is also not clear, why 3 different FTIR image was used. Author should add a FTIR of 400-3400 Cm-1 range and also define all the FTIR peak in Figure.

Author reply à According to the reviewer suggestion, a FTIR in the range 400-3400 cm-1 is now provided as Fig. 6b while Fig. 6c represents a zoom in the region 400-3400 cm-1, evidencing a peak relative to the Zeo-sal filler (C-H aromatic).

5.         What about the biocompatibility of the developed composite material?

Author reply à The aim of the paper was the possibility to prepare composites based on food grade zeolite as carrier of an antimicrobial molecule of interest in food packaging. The biocompatibility of the systems is out of the aim of this work.

6.        What is the application of the prepared composite?  Here sodium salicylate was used as an antimicrobial agent so where is antimicrobial activity test results?

Author reply à It has been reported the effect of sodium salicylate on bacteria (https://doi.org/10.1016/S1357-2725(00)00042-X). The evaluation of the antimicrobial activity against selected bacteria responsible of food spoilage is a work in progress.  

7.       What about the mechanical properties of the prepared composite film?

Author reply à We evaluated the mechanical properties of the composites and, for comparison, of PA11. Results are reported in a novel figure (Figure 7).

8.         The discussion is too short (Line 311), expand it

Author reply à The discussion is provided within the description of the results.

9.       In line 311, Author claimed nanocomposite but the results do not support the same. Give a proper explanation.  

Author reply à We used in the whole text the word “composites” instead of “nanocomposites”.

10.         The conclusion is missing, add a brief conclusion at the end of the manuscript.

Author reply à We wrote Conclusions, reporting the most important results, at the end of the manuscript.

11.       Also, carefully revise the manuscript for typographical and linguistic error.

Author reply à We revised the whole manuscript.

We add the revised manuscript with highlighted in yellow our corrections/modifications according to Reviewer's suggestions

Round 2

Reviewer 2 Report

The article can be published after revision being made according to the suggestion given below:

Check line 36, 37-38.

The surface SEM image (Fig. 3) is still not clear.

The results and discussion part did not improve. Need more revision.

I could not find any expansion of discussion. Revise it carefully.

Author Response

Dear Editor,

I send you the second revised version of the original paper “Ball Milling to produce composites based of natural clinoptilolite as carrier of salicylate in bio-based PA11” by Valeria Bugatti, Paola Bernardo, Gabriele Clarizia, Gianluca Viscusi, Luigi Vertuccio and myself to be considered for publication in Polymers (Special Issue "Materials and Methods for New Technologies in Polymer Processing").

We thank again the Reviewer #2 for his/her further suggestions that greatly helped to improve the quality of the paper. The yellow text evidences our modifications.

Below you find our point by point answer.

We are confident that the manuscript will now be suitable for publication in Polymers.

I thank you for your time and concern and I send you my best regards,

Giuliana Gorrasi

prof. Giuliana Gorrasi

Department of Industrial Engineering-University of Salerno-

via Giovanni Paolo II 132, 84084 Fisciano (SA)-Italy

e-mail: ggorrasi@unisa.it

tel: +39089964146-4019; fax: +39089964057

Answer to Reviewer #2

Check line 36, 37-38.

Done

The surface SEM image (Fig. 3) is still not clear.

New SEM images of the surface were taken

The results and discussion part did not improve. Need more revision.

I could not find any expansion of discussion. Revise it carefully.

We improved the discussion part
